# G Protein-Coupled Receptor 37L1 Modulates Epigenetic Changes in Human Renal Proximal Tubule Cells

**DOI:** 10.3390/ijms232214456

**Published:** 2022-11-21

**Authors:** Ines Armando, Santiago Cuevas, Caini Fan, Megha Kumar, Zahra Izzi, Pedro A. Jose, Prasad R. Konkalmatt

**Affiliations:** Division of Renal Diseases & Hypertension, Department of Medicine, The George Washington University, Washington, DC 20037, USA

**Keywords:** blood pressure, DNMT1, epigenetics, ETBR-LP2, G protein-coupled receptor, GPCR, GPR37L1, hypertension, mass spectrometry, NHE3 signaling pathway, PI3/AKT/mTOR, renal proximal tubule, sodium transport

## Abstract

Renal luminal sodium transport is essential for physiological blood pressure control, and abnormalities in this process are strongly implicated in the pathogenesis of essential hypertension. Renal G protein-coupled receptors (GPCRs) are critical for the regulation of the reabsorption of essential nutrients, ions, and water from the glomerular filtrate. Recently, we showed that GPCR 37L1 (GPR37L1) is expressed on the apical membrane of renal proximal tubules (RPT) and regulates luminal sodium transport and blood pressure by modulating the function of the sodium proton exchanger 3 (NHE3). However, little is known about GPR37L1 intracellular signaling. Here, we show that GPR37L1 is localized to the nuclear membrane, in addition to the plasma membrane in human RPT cells. Furthermore, GPR37L1 signals via the PI3K/AKT/mTOR pathway to decrease the expression of DNA (cytosine-5)-methyltransferase 1 (DNMT1) and enhance NHE3 transcription. Overall, we demonstrate the direct role of a nuclear membrane GPCR in the regulation of renal sodium through epigenetic gene regulation.

## 1. Introduction

Hypertension affects one-third of the U.S. adult population and is not well controlled in approximately half of adults with hypertension [1]. Hypertension is a complex polygenic multifactorial disorder, but only <5% of the genetic factors that are linked to blood pressure (BP) regulation have been identified [2]. The understanding of the mechanisms by which individual genes and gene modifiers participate in the pathogenesis of diseases [3], including hypertension, is essential to develop drugs that control hypertension better and with fewer side effects. Aberrations in renal epithelial ion transport contribute to the pathogenesis of essential hypertension [4]. The human genome codes for over 800 G protein-coupled receptors (GPCRs) which regulate almost all aspects of cellular physiology [5,6], and GPCRs in the kidney regulate the reabsorption of essential nutrients, ions, and water from the glomerular filtrate [4,7]. The number of identified GPCRs that are expressed in the kidney continues to increase, but the functional role and regulation of the expression of several of these GPCRs have yet to be explored [8,9,10]. Therefore, elucidating the roles of renal GPCRs in both healthy and disease states may shed important new light on the regulation of BP and the pathophysiological mechanisms underlying hypertension.

In humans and mice, *GPR37L1* is located in chromosome 1, 1q32.1 and 1E4, respectively, and in chromosome 13q13 in rats. Renin is the first BP quantitative trait locus (QTL) identified by genetic linkage analyses [11]. Later, Flister et al., in 2013, mapped two loci close to renin, one that decreased the BP and an adjacent ~1 MB loci that increased BP and albuminuria [12]. *GPR37L1* is located within this 1 MB hypertensive loci region. Genomic organization within this 1 MB hypertensive locus is highly similar among rat, mouse, and human, except for LncRNA within the human GPR37L1 and a Gm41955 between ELF3 and GPR37L1 in mouse (Appendix A).

The function of *Gpr37l1* in neuronal astrocytes is well studied. Global knockout of *Gpr37l1* is associated with high susceptibility to seizures [13], increased neuronal loss following stroke [14], and delayed onset of tumor in a mouse model of medulloblastoma [15]. However, the function of GPR37L1 in peripheral organs is not well studied. Previous reports on the role GPR37L1 on BP regulation are inconclusive due to a lack of detailed analyses. Arking et al. (2010) reported a single nucleotide polymorphism (SNP, rs7516762) near the *GPR37L1* gene in a genome-wide association study (*n* = 424, and 226 controls) on sudden cardiac arrest in patients with coronary artery disease [16]. The association of this SNP with cardiac disease has not been validated. Min et al., in 2010, performed a microarray analysis on heart explants from patients with heart failure associated with high pulmonary arterial pressure and low ejection fraction [17]. *GPR37L1* was one of the 12 genes that were analyzed further based on its decreased expression in the heart of these patients. They found that in cardiac myocytes, overexpression of *Gpr37l1* with alpha-myosin heavy chain promoter [18] have decreased viability and increased apoptosis. BP of transgenic *Gpr37l1* mice was lower than in wild-type mice. The germline deletion of *Gpr37l1* caused cardiac hypertrophy (increase in heart weight to body weight ratio) and increased BP in two reports, but only in female mice in one of these reports [19,20]. However, germline deletion of *Gpr37l1* has also been reported to decrease BP in female but not in male mice [20]. We recently reported that GPR37L1 is expressed in the kidney at the apical membrane of renal proximal tubules (RPT) and human renal proximal tubule cells (hRPTCs). Renal-specific silencing of *Gpr37l1* in mice increased renal sodium excretion and decreased BP [21]. Although our results on the effect of GPR37L1 on BP appear to contradict the report by Min et al., Coleman et al., and Mouat et al. [17,19,20], it should be noted that the studies by these groups showed the effect of global knockout of *Gpr37l1*, whereas we showed the effect on BP of specific silencing of *Gpr37l1* in the kidney. In hRPTCs, overexpression of GPR37L1 increases luminal sodium transport by increasing NHE3 expression and/or function [21].

The ability of GPR37L1 to increase sodium transport in hRPTCs was associated with a decrease in intracellular cAMP production, similar to previous reports indicating binding to Gαi protein [22]. GPR37L1 has been shown to have constitutive activity [23]. However, in the cerebellum, the putative GPR37L1 ligand, prosaposin, does not affect Gα_S_ or Gα_i_ signaling [24]. GPR37L1 signaling may also involve the phosphorylation of extracellular signal-regulated kinases in astrocytes [25]. These studies indicate that the GPR37L1 signaling pathway is not clearly understood, especially outside the nervous system.

In classical GPCR signaling, stimulation with an agonist triggers dissociation of heterotrimeric G proteins, inhibition or stimulation of downstream effector molecules, and internalization of the receptor, causing downregulation of the signaling pathway [26]. Several recent studies have shown that GPCRs can also signal at intracellular membranes such as endosomes, Golgi compartments, and nuclear membranes, as well as the nucleus, and that the GPCRs continue to be activated in these compartments [27,28]. A growing number of GPCRs and their interacting protein components have been shown to be localized on the nuclear membrane [29,30] or within the nucleus [31,32], where they retain their signaling function [33,34,35,36]. Ligand-induced translocation of GPCRs to the nucleus has been shown for GPR158 [37], angiotensin II receptor 1 (AT1R) [38], oxytocin receptor (OTR) [39], and other receptors [40]. Localization at the nuclear membrane or the nucleus occurs via a classical nuclear localization sequence (NLS) and/or importins [28,30].

In this report, we show that GPR37L1 is expressed at the plasma and nuclear membranes, signals via the ERK1/2/PI3K/AKT/mTOR (mammalian target of rapamycin) pathway, and modulates NHE3 expression through DNA (cytosine-5)-methyltransferase 1 (DNMT1)-dependent methylation of the NHE3 promoter. These results provide the mechanism that supports our previous report showing the ability of GPR37L1 to increase renal luminal sodium transport by increasing the expression of NHE3 [21].

## 2. Results

### 2.1. GPR37L1 Is Associated with Components of PI3K/AKT Signaling Pathway and Nuclear Transport Proteins

We employed affinity purification and mass spectrometry to identify the proteins that directly interact with GPR37L1. The proteins from human renal proximal tubule cells (hRPTCs) expressing GPR37L1-GFP were fractionated by glycerol gradient centrifugation (Figure 1). Immunoblotting of the glycerol gradient fractions showed most of the GPR37L1-GFP distributed in fractions 10–14 (Figure 1B), suggesting that GPR37L1 exists as a protein complex in hRPTCs. To identify the components of the GPR37L1 protein complex, fractions 10–14 (Figure 1B, red line) were pooled and the GPR37L1 protein complex was captured by immunoaffinity purification, using anti-GFP antibodies. Mass spectrometry (MS) analyses of the immunocomplex revealed the association of GPR37L1 with kinases involved in signaling, such as mTOR, insulin receptor substrate 4 (IRS4), protein kinase DNA-activated catalytic subunit (PRKDC); DNMT1, a DNA methylation enzyme, nuclear import proteins IPO1 and IPO7, nuclear export proteins XPO1 and XPO5; nuclear pore complex proteins such as importin subunit beta-1 (KPNB1), importin subunit alpha-1 (KPNA2), and a nuclear pore membrane glycoprotein 210 (NUP210O) (Table 1). The interaction of GPR37L1 with several proteins that participate in importing and exporting proteins in and out of the nucleus and with a nuclear pore membrane suggests that GPR37L1 may also be localized to the nucleus.

### 2.2. GPR37L1 Is Expressed on the Nuclear Membrane

To determine the location of GPR37L1 expression, we performed fluorescence imaging of hRPTCs expressing GPR37L1-GFP. Confocal images of the hRPTCs showed expression of GPR37L1-GFP on the plasma membrane (Figure 2A, white arrow), and expression around the nucleus (Figure 2A, red arrows). The fluorescence imaging of nuclei prepared from the cells expressing GPR37L1-GFP showed that the expression is predominantly on the nuclear membrane (Figure 2B, red arrows). Very little or no GPR37L1-GFP signals were detectable inside the nucleus. These results show that the GPR37L1 is expressed on the nuclear membrane.

As mentioned above, several GPCRs expressed in the nuclear membrane or in the nucleus contain a nuclear localization signal (NLS) [28,39,40]. In silico amino acid sequence analyses of GPR37L1 using the nuclear localization signal predictor tools (http://nls-mapper.iab.keio.ac.jp/cgi-bin/NLS_Mapper_form.cgi (accessed on 15 March 2019) and http://mleg.cse.sc.edu/seqNLS/ (accessed on 15 March 2019)) revealed the presence of a potential bipartite and classical monopartite NLS at the amino terminus (Figure 3A). Next, we evaluated if the predicted NLS on the N-terminus region of GPR37L1 is required for its nuclear localization. Total or nuclear proteins prepared from hRPTCs transfected with a GPR37L1-GFP plasmid were immunoblotted to detect GPR37L1-GFP. The results show that a ~75 kDa GPR37L1-GFP is present in the total, as well as in the nuclear protein preparations (Figure 3B). To assess the purity of the nuclear protein preparation, the same blot was sequentially probed for calnexin (Figure 3C), a marker for endoplasmic reticulum (ER); Na^+^/K^+^-ATPase (Figure 3D), a marker for plasma membrane; and HDAC2 (Figure 3E), a marker for the nucleus. The results show that calnexin and Na^+^/K^+^-ATPase were absent in the nuclear protein preparation but present in the total protein preparations. HDAC2 was present in the total, as well as in the nuclear protein preparations. These results confirm that the nuclear protein preparation is devoid of ER and plasma membrane proteins. Furthermore, these results confirm that the integral plasma membrane protein GPR37L1 is also localized to the nuclear membrane of hRPTCs.

### 2.3. NLS Is on the N-Terminus of GPR37L1

Next, we evaluated if the predicted NLS on the N-terminus region of GPR37L1 is required for its nuclear localization. We prepared constructs in which the N-terminus 124-amino-acid region or the predicted NLS in the amino terminus of GPR37L1 is tagged to a luciferase and GFP (LG) fusion protein (Figure 4A). Locations of the expressions of N-terminus or NLS-fused LG constructs were determined in hRPTCs by immunoblotting and immunofluorescence.

Immunoblot analyses of the N-terminus and NLS constructs showed the expressions of predicted protein sizes (85 kDa from 37L1-Nter-LG and 75 kDa from 37L1-WtNLS-LG) in the total, as well as in the nuclear protein preparations (Figure 4B). These results suggest that the predicted 13-amino-acid sequence is adequate for nuclear expression of GPR37L1. Further mutations on the predicted 13-amino-acid NLS that converted the basic amino acid sequence RSRK to neutral amino acids GSAA also produced a predicted 75 kDa protein observed in the total and nuclear protein preparations (Figure 4B). In the total protein preparations, there was no significant difference between the levels of expressions of WT- or Mu-NLS constructs. In the nuclear protein preparations, the expression of the Mu-NLS was markedly decreased compared with that of Wt-NLS (Figure 4B and Appendix A). Nuclear to total expression ratio was (74 +/−8%) lower in the Mu-NLS than the Wt-NLS (Appendix A), while sequential immunoblotting for HDAC2 showed equal protein loading of the nuclear proteins from Wt- and Mu-NLS constructs. The same blot was sequentially probed for GAPDH to demonstrate that the nuclear protein preparation was free from cytoplasmic proteins (Figure 4C). GAPDH was present in the total protein preparations, but not in any of the nuclear protein preparations. Therefore, the basic amino acid sequence RSKR at the amino terminus is crucial for nuclear expression of GPR37L1. The persistence of some nuclear expression of mutant NLS, i.e., Mu-NLS construct, suggests that in addition to RSKR, there may be other regions contributing to the nuclear expression of GPR37L1.

Next, we performed immunofluorescence analyses to determine whether the amino terminus RSKR is essential for GPR37L1 expression in the nuclear membrane. Fluorescence imaging showed that a full-length GPR37L1 tagged with GFP (GPR37L1-GFP) colocalized with lamin B1, a marker for the nuclear inner membrane, demonstrating the expression of GPR37L1 in the nuclear membrane (Figure 5). Confocal images of hRPTCs expressing 37L1-WtNLS-LG showed diffuse expression within the nucleus, as well as in the nuclear membrane (white areas [green + blue + red], Figure 5) colocalizing with lamin B1. The expression of 37L1-MuNLS-LG was also observed in the nucleus as well as on the nuclear membrane, but the expression in the nucleus or in the nuclear membrane was not as pronounced as that observed with the 37L1-WtNLS-LG (Figure 5), corroborating their quantified expressions in Figure 4C. The expression of GFP alone was observed throughout the cell, including in the nucleus. These results confirm that the amino acid sequence RSKR at the amino terminus of GPR37L1 is necessary for its nuclear expression.

### 2.4. GPR37L1 Regulates the Expression of DNMT1 through ERK1/2/PI3K/AKT/mTOR Pathway

MS analyses of the GPR37L1 immunocomplex (Table 1) revealed the association of GPR37L1 with DNMT1, mTOR, and PRKDC. To confirm the association of GPR37L1 with these proteins, we performed coimmunoprecipitation (Co-IP) of GPR37L1 proteins complex fractionated over a glycerol gradient with GFP antibody and immunoblotted with the antibodies for GPF37L1, DNMT1, mTOR, or PRKDC (Figure 6). The GPR37L1-GFP-specific immunoreactive band corresponding to ~75 kDa was observed only in the protein preparations of GPR37L1-GFP expressing hRPTCs co-immunoprecipitated with the GFP antibody, but not with the control IgG (Figure 6). GPR37L1-GFP-specific immunoreactive bands were absent in the Co-IP of protein preparation from mock-transfected cells using control IgG or GFP antibody. DNMT1-specific immune reactive bands corresponding to a predicted 200 kDa were observed in the protein preparations from hRPTCs expressing GPR37L1-GFP, but not with the control IgG. DNMT1-specific immunoreactive bands were absent in the Co-IP of protein preparation from mock-transfected cells using control IgG or GFP antibody (Figure 6). We also found mTOR (~280 kDa) in the GPR37L1-GFP and GFP co-IP (Figure 6). Similarly, immunoblotting for PRKDC showed a ~350 kDa PRKDC-specific immunoreactive band (Figure 6) in the protein preparations from GPR37L1-GFP expressing hRPTCs co-immunoprecipitated with the GFP antibody, but not with the control IgG (Figure 6).

One of the main downstream targets of the PI3K/AKT signaling pathway is mTOR. mTOR is an atypical serine/threonine kinase that forms two distinct complexes. The first, mTOR complex 1 (mTORC1), which is inhibited by rapamycin, regulates gene transcription and protein synthesis to regulate cell growth and proliferation by phosphorylating substrates that potentiate anabolic processes or limit catabolic processes. The second complex, mTOR complex 2 (mTORC2), acts as a kinase phosphorylating AKT at S473 in cancer and adipose cell lines [41,42,43,44]. PRKDC, a member of PI3K family serine/threonine kinase, can phosphorylate AKT at S473, following DNA damage or in the presence of CpG DNA [45,46,47,48].

Several GPCRs not only signal through the Gα protein to which they bind but also signal through Gβγ to activate ERK1/2 signaling, which in turn activates PI3K [49]. These observations put together with our findings suggest that GPR37L1 may signal via PI3K/AKT/mTOR together with PRKDC. Therefore, we probed the ERK1/2/PI3K/AKT pathway in hRPTCs transfected with GPR37L1 plasmid.

The transient transfection with the GPR37L1 plasmid increased the expression of GPR37L1 (8.7 ± 0.5-fold), compared to mock-transfected cells (Figure 7A). The expression of GPR37L1 increased the phosphorylation of ERK1/2, PI3K, and AKT at S473 (Figure 7B–D), confirming that GPR37L1 signals through ERK1/2/PI3K/AKT.

In vascular endothelial cells, inhibition of mTOR by rapamycin decreases the expression of DNMT1 [50], while in hepatocellular carcinoma cell lines Hep G2, SNU-182, and Hep 3B2.1-7, inhibition of mTOR with Torin-2 increases the expression of DNMT1 [51]. We found that the overexpression of GPR37L1 decreased the expression of DNMT1 by 70 ± 8% (Figure 8A) and rapamycin abolished the ability of GPR37L1 to reduce DNMT1 expression (Figure 8B). To confirm our findings, we studied hRPTCs endogenously expressing GPR37L1; treatment with rapamycin increased the expression of DNMT1 in a dose-dependent fashion in these cells (Figure 9A) and increased the expression of NHE3 (Figure 9B). Therefore, GPR37L1 regulates DNMT1 expression via the ERK1/2/PI3K/AKT/mTOR pathway and may participate in epigenetic gene regulation of NHE3 by modulating DNA methylation.

We hypothesized that the decreased expression of DNMT1 in GPR37L1-expressing cells increases NHE3 expression by reducing CpG methylation on its promoter. Global DNA methylation assay on hRPTCs expressing GPR37L1 showed a trend to reduce 5-cytosine methylation. However, the reduction in methylation did not reach statistical significance. This could be taken to indicate that the GPR37L1 effect on DNA methylation is gene-specific. Indeed, rapamycin treatment had no effect on the methylation status in mock-transfected controls but increased it in GPR37L1-expressing cells (Figure 10A). Methylated DNA immunoprecipitation (Me-DIP) using 5-methyl cytosine-specific antibody (5mc-Ab) followed by quantitative PCR analyses of NHE3 promoter (region −463 to −342 from the TSS) showed a 58 ± 11% reduction in the methylated NHE3 promoter in the GPR37L1-expressing cells compared with mock-transfected control cells (Figure 10B, Veh-treated, Mock vs. GPR37L1-transfected). However, rapamycin treatment reversed the inhibitory effect of GPR37L1 on the methylation of NHE3 promoter (Figure 10B, GPR37L1-transfected, Veh vs. Rapamycin-treated). In the mock-transfected cells, rapamycin treatment increased NHE3 promoter methylation by 26 ± 4.5% compared with the vehicle-treated cells (Figure 10B, Mock-transfected, Veh vs. Rapamycin-treated). Collectively, our data show that in hRPTCs, GPR37L1 signals through the ERK1/2/PI3K/AKT/mTOR pathway to decrease DNMT1 expression and consequently the methylation status of the NHE3 promoter, positively regulating the expression of NHE3.

## 3. Discussion

We previously reported that GPR37L1 increases renal luminal sodium transport by increasing the expression of NHE3 [21]. Here, we show that GPR37L1 increases renal NHE3 expression by the methylation of its promoter region through the ERK1/2/PI3/AKT/mTOR pathway. This is the first report showing the direct role of a GPCR in the regulation of renal sodium transport through an epigenetic gene pathway. GPCRs expressed in the renal proximal tubule play a critical role in sodium reabsorption and maintenance of BP, and hence are the targets for pharmaceutical intervention to treat hypertension [6,52,53,54]. A growing number of GPCRs are expressed in the nephron [10,21]. However, several GPCRs have not gained attention, either because their functions have yet to be linked to known clinical conditions or their binding partners have not been identified. We have shown that GPR37L1 is expressed on the brush border of hRPTCs and participates in increasing luminal sodium reabsorption and BP [21].

Detection of GPCR expression in the nuclear membrane changed the conventional notion that GPCRs signal solely from the plasma membrane [27,28,29,30,31,32,33]. To date, approximately 40 GPCRs are known to be expressed in the nucleus or in the nuclear membrane. Ligand-mediated translocation of GPCRs from the plasma membrane to the nucleus has been demonstrated for several receptors [34,35,36,37,38,39,40]. Most nuclear GPCRs are constitutively expressed in the nuclear membrane through their intrinsic classical monopartite or bipartite NLS [28]. Our present results demonstrate that GPR37L1 is one of the few GPCRs expressed in the nuclear membrane. The nuclear membrane expression of GPR37L1 occurs mainly via the NLS located in the extracellular N-terminus, mutations of which markedly decreased, but did not abolish, the nuclear expression of GPR37L1. This suggests that there may be other intrinsic sequences within the GPR37L1 involved in its nuclear membrane expression.

We have reported that GPR37L1 increased luminal sodium transport in hRPTCs by decreasing cAMP production and increasing the expression of NHE3 [21]. However, the precise GPR37L1 signaling pathway was not studied. We now report the association, confirmed by Co-IP, of GPR37L1 with the kinases mTOR, PRKDC, and DNMT1. As mentioned above, mTOR is both a main downstream target of pAKT and also a kinase that can phosphorylate AKT [41]. PRKDC is a member of PI3K family serine/threonine kinase and is also shown to phosphorylate AKT following DNA damage or in the presence of CpG DNA [47,48].

We and others showed that GPR37L1 decreases cAMP production, indicating that it is coupled with Gαi [21,22]. These results together with previous reports [49] indicate that GPR37L1 signals via the Gβγ-mediated ERK1/2/PI3/AKT/mTOR pathway. DNA methyltransferases catalyze the transfer of a methyl group from S-adenosyl-L-methionine to the fifth position of cytosine in the cytosine guanine dinucleotides (CpG) in the genomic DNA. Five such enzymes have been identified: DNMT1, DNMT2, DNMT3A, DMNT3B, and DNMT3. DNMT1 is shown to methylate the DNA during the replications, whereas DNMT3A and DNMT3B are known to maintain the methylation of genomic DNA [55,56].

Inhibition of mTOR by rapamycin either increases or decreases the expression of DNMT1 depending on the cell type studied [50,51]. We found that in hRPTCs, GPR37L1 decreased DNMT1 expression, whereas rapamycin inhibition of mTOR increased DNMT1 expression in a concentration-dependent fashion, which is consistent with an earlier report.

Our results showed that GPR37L1 may participate in epigenetic gene regulation by decreasing the methylation status of genomic DNA. However, GPR37L1 only tended to reduce 5-cytosine methylation, which could indicate that that the GPR37L1 effect on DNA methylation is gene-specific, e.g., Nhr3 (vide infra). NHE3 is expressed on the apical membrane of the renal proximal tubule, which is responsible for about 60% of the sodium reabsorption by the proximal tubule [57,58,59].

In silico analyses of the human NHE3 promoter using http://www.urogene.org/cgi-bin/methprimer/methprimer.cgi (accessed on 18 October 2022) predicted a dense CpG island in the ~400 bp region upstream of the transcription start site in human NHE3 promoter. Intestinal epithelial cell lines Caco-2 and HuTu 80 treated with 5-azacytidine, an inhibitor of DNA methyltransferase, have increased the expression of NHE3 mRNA, while inhibition of mTOR with rapamycin decreased the expression of NHE3 in mouse and human intestines [60,61]. Our studies indicate that the expression of NHE3 is negatively regulated by DNA methylation. MeDIP assay of the NHE3 promoter showed that the expression of GPR37L1 reduced the methylation on the NHE3 promoter, which would increase NHE3 transcription [21]. In mock-transfected cells, rapamycin treatment increased the level of methylated NHE3 promoter and abolished the inhibitory effect of GPR37L1 on the NHE3 promoter methylation.

Our data showed that in RPTCs, GPR37L1 signaled via the ERK1/2/PI3K/AKT/mTOR pathway and increased the transcription of NHE3 through a DNMT1-mediated decrease in the methylation of the NHE3 promoter. The integrated proposed pathway of GPR37L1 effect on NHE3 expression from the plasma membrane to the nucleus is shown in Figure 11.

This is the first report showing the direct role of a GPCR in the regulation of renal sodium transport through epigenetic gene regulation. These results support our earlier report showing that GPR37L1 increases renal luminal sodium transport by increasing the expression of NHE3 [21].

## 4. Materials and Methods

### 4.1. Compounds

The drugs used in the present study were purchased from Sigma-Aldrich (St. Louis, MO, USA), unless indicated otherwise: rapamycin (Cell Signaling Technology, Boston, MA, USA); plasmocin (Fisher Scientific, Hampton, NH, USA); EGF, dexamethasone, triiodothyronine, insulin–transferrin–selenium (ITS, 1X, Invitrogen, Carlsbad, CA, USA); penicillin/streptomycin (1X, Invitrogen); fetal bovine serum (Invitrogen); L-glutamine, HEPES, sodium pyruvate, and 2-mercaptoethanol. Trypsin-ultra, mass spectrometry-grade, restriction enzymes, and other molecular biology reagents were purchased from New England BioLabs (Ipswich, MA, USA).

### 4.2. Cell Culture

Immortalized hRPTCs isolated from normotensive Caucasian males (gift form Dr. Robin A. Felder, University of Virginia, Charlottesville, VA, USA) [62] were grown in DMEM/F12 supplemented with 2% fetal bovine serum (FBS, epidermal growth factor (EGF, 10 ng/mL), an insulin–transferrin–selenium cocktail (5 μg/mL each), and dexamethasone (4 ng/mL) in a 37 °C incubator with humidified 5% CO_2_ and 95% air.

### 4.3. Plasmids

A schematic diagram of the plasmids used is shown in Figure 2A. A plasmid encoding human GPR37L1 (p37L1) or GPR37L1 tagged with turbo green fluorescence protein (GFP) (p37L1-GFP) was obtained from Origene Technology Inc. (Rockville, MD, USA; cat. # SC117166 and RG208132, respectively). For construction of 37L1-Nter-LG, first, a mutant containing the N-terminus 124 amino-acids of GPR37L1 tagged to GFP (37L1-Nter-GFP) was generated by removing the regions that code for the amino-acid sequence 133–481 from p37L1-GFP by restriction digestion. Next, the mutant carrying 124 amino-acid N-terminus of GPR37L1 tagged with luciferase–GFP chimeric cDNA (pNter-LG) was generated by the in-frame insertion of a PCR-amplified firefly luciferase coding sequence without start codon ATG and stop codon TAA, between the N-terminus of GPR37L1 and GFP coding sequence in 37L1-Nter-GFP, so that the resulting plasmid (37L1-Nter-LG) expresses fused luciferase–GFP fused protein tagged with N-terminus of GPR37L1. The plasmids 37L1-WtNLS-LG or 37L1-MuNLS-LG were generated by replacing the coding sequence for the predicted wild-type (WT) or mutant (Mu) monopartite NLS: QQSRSKRGTEDEE or QQSGSAAGTADEE in place of the N-terminus coding sequence in the plasmid 37L1-Nter-LG.

### 4.4. Glycerol Gradient Fractionation and Mass Spectrometry

hRPTCs were transfected with the plasmids p37L1-GFP. Two days later, total cellular protein preparations were subjected to glycerol gradient centrifugation and mass spectrometry analyses, as described previously [63,64,65,66]. Briefly, cells from one 15 cm plate were resuspended with 950 μL of lysis buffer (50 mM HEPES pH 7.6, 150 mM KCl, 2 mM EDTA, 0.5 mM DTT, and protease (Sigma P8340) phosphatase (Calbiochem 524629)). The cells were lysed with the addition of NP40 (10%) to a final concentration of 0.5%, agitated gently at 4 °C for 15 min, and clarified by centrifugation at 10,000× *g* for 15 min. The soluble protein samples were fractionated over 10–30 glycerol gradient centrifugation at 175,000× *g* for 5 h. Then, 0.5 mL fractions were collected from the top of the gradient. The fractions containing the GPR37L1 protein complex, identified by immunoblotting with GFP-specific antibody, were pooled and immunoaffinity-purified using the GFP antibody immobilized on Protein G-coated Dynabeads (10009D, Dynabeads™ Protein G for Immunoprecipitation, Invitrogen, Carlsbad, CA, USA). The proteins from Dynabeads were eluted with SDS sample buffer and loaded onto the SDS-PAGE and stained using Bio-Safe Coomassie Stain (BioRad).

The immunoaffinity purified proteins samples were loaded on the 10% SDS-PAGE and electrophoresed until the migration just entered the separating gel. The proteins were extracted following in-gel digestion with trypsin, as described previously [66]. The resulting peptides were extracted and analyzed by liquid chromatography–tandem mass spectrometry.

### 4.5. Mass Spectrometry Analyses

In order to determine the protein binding partners of GPR37L1, immunoprecipitation–mass spectrometry (IP-MS) experiments were performed. The cell lysates of hRPTCs, transfected with the plasmids p37L1-GFP, were centrifuged through a glycerol gradient and collected in 16 fractions. The fractions were immunoblotted with GPR37L1 antibody and fractions 10–14 were selected for IP-MS analysis. IP was performed using anti-GPR37L1 antibodies on each pooled sample. The IP lysates were collected for MS analysis.

Using IP-MS standard protocols [67], the protein fractions were prepared for and analyzed by LC-MS/MS using a nano-LC system (Easy nLC1000) connected to a Q Exactive HF mass spectrometer (Thermo Scientific, Waltham, MA, USA). The MS raw datasets, which included common contaminants, were searched against the UniProt Mouse database using MaxQuant software (version 1.5.5.1). After “sticky” proteins [68] were removed, a total of 172 proteins were identified by IP-MS in two independent experiments (*n* = 2 experiments). The presence of the protein DNA (cytosine-5)-methyltransferase 1 (DNMT1), mammalian target of rapamycin (mTOR), and DNA-dependent protein kinase (PRKDC, also known as DNA-PK) suggested the physical interaction of these proteins with GPR37L1.

### 4.6. Total and Nuclear Protein Preparation

hRPTCs from 10 cm plates were collected in 5 mL cold PBS by scraping and centrifuged at 100× *g* for 5 min. For total protein preparation, the cell pellets were resuspended in RIPA buffer, containing protease and phosphatase inhibitor. For nuclear protein preparation, the cell pellets were resuspended in 0.5 mL STM-NP buffer (10 mM Tris-HCl pH 8.0, 0.25 M sucrose, 10 mM MgCl2, 1 mM DTT) with 0.5% NP-40 and protease and phosphatase inhibitors, and mixed gently at 4 °C for 10 min. The nuclei were pelleted by centrifugation at 1000× *g* for 5 min. The pelleted nuclei were washed four times with 1 mL of STM-NP buffer without NP-40. The nuclei were resuspended in SDS sample buffer. The cell and nuclei lysates were sonicated and clarified by centrifugation at 10,000× *g* for 10 min. Protein content was determined using the *DC™* Protein Assay kit (BIO-RAD, Hercules, CA, USA)

### 4.7. In Silico Analyses

Methylation-specific (M) and unmethylation-specific (U) primer sequences for PCR amplification of the NHE3 promoter were designed using the http://www.urogene.org/cgi-bin/methprimer2/MethPrimer.cgi (accessed on 18 October 2022). Nuclear localization signal predictor tools http://nls-mapper.iab.keio.ac.jp/cgi-bin/NLS_Mapper_form.cgi (accessed on 18 October 2022) and http://mleg.cse.sc.edu/seqNLS/ (accessed on 18 October 2022) were used to identify potential bipartite and classical monopartite NLS on the amino terminus of GPR37L1.

### 4.8. Immunoblotting

hRPTCs were homogenized in RIPA buffer with protease and phosphatase inhibitors, and equal amounts of protein (20 μg) were electrophoresed under reducing conditions on polyacrylamide gels and then transferred onto nitrocellulose membranes. Target proteins were detected by a standard protocol, using target protein or phospho-protein-specific first antibody and an appropriate second antibody. The primary antibodies used were anti-GPR37L1 (catalog# sc-164532, Santa Cruz Biotechnology, Santa Cruz, CA, USA), anti-PRKDC (catalog#38169, Cell Signaling Technology, Danvers, MA, USA), anti-phospho ERK1/2 (catalog#4370, Cell Signaling), anti-ERK1/2 (catalog#4695 Cell Signaling), anti-phospho-AKT (catalog#9271, Cell Signaling Technology), anti-AKT (catalog#4691, Cell Signaling Technology), anti-PI3K (catalog#4257, Cell Signaling Technology), anti-phospho-PI3K (catalog#4228, Cell Signaling Technology), anti-PP2A (catalog#2259, Cell Signaling Technology), anti-DNMT1 (catalog#5032, Cell Signaling Technology), anti-mTOR (catalog#2983, Cell Signaling Technology), anti-HDAC2 (catalog#5113, Cell Signaling Technology), normal rabbit IgG (catalog#, Cell Signaling Technology), anti-lamin B1 (catalog# MAB8525SP, R&D Systems, Minneapolis, MN, USA), anti-mC-mAb (catalog#A-1014-050, Epigentek, Farmingdale, NY, USA), anti-Na^+^/K^+^-ATPase (catalog# A276, Sigma-Aldrich, St. Louis, MO, USA), anti-GAPDH(catalog#AB2302, Sigma-Aldrich, St. Louis, MO, USA), anti-tubulin (catalog#, T9026, Abcam, Cambridge, MA, USA), anti-calnexin (catalog# 2679, Cell Signaling), anti-turboGFP (catalog# 2983), and anti-NHE3 (catalog# NBP3-00589, Novus Biologicals, Centennial, CO, USA). The fluorescence-tagged second antibodies were IRDye^®^ 800CW-labeled donkey anti-rabbit IgG (926-32213, Licor, Lincoln, NE, USA) or IRDye^®^ 680RD-labeled donkey anti-mouse IgG (926-68022, Licor, Lincoln, NE, USA). Results were normalized to GAPDH or tubulin and expressed as fold-change, relative to the average signal intensity obtained from the respective controls.

### 4.9. Immunofluorescence

Sub-cellular locations of GFP-tagged GPR37L1 wild-type or the constructs carrying N-terminus or NLS of GPR37L1 tagged to luciferase–GFP were documented by immunofluorescence analysis. The hRPTCs transfected with the GPR37L1 constructs were fixed in 3.7% paraformaldehyde for 10 min. After washing, the cells were immunostained for GFP using anti-GFP antibody. Where indicated, the cells were also double-stained for lamin B1, a marker for inner membrane of the nucleus. Cells were also stained with 4′,6-diamidino-2-phenylindole (DAPI) to mark the nuclei. Fluorescence images were captured using the Zeiss LSM 710 laser scanning confocal microscope.

### 4.10. Quantitative Real-Time PCR

Total RNA was prepared from hRPTCs using the RNeasy Plus Mini kit (Qiagen, Hilden, Germany). cDNA was prepared using an RT2 First Strand kit, as per the manufacturer’s protocol (SABiosciences-Qiagen). Quantitative gene expression was analyzed by real-time qPCR using an ABI Prism 7900 HT (Applied Biosystems, Waltham, MA, USA). RT2 SYBR Green ROX qPCR Master Mix (Qiagen Inc.) was used. Target gene-specific primers were obtained from Integrated DNA technologies Inc. (Coralville, IA, USA). Data were analyzed using the ΔΔCt method [69].

### 4.11. Global DNA Methylation Assay

Genomic DNA was prepared using DNeasy Blood and Tissue Kit (Qiagen, Fremont, CA, USA) from the hRPTCs expressing GPR37L1, following the manufacturer’s protocol. DNA methylation status was quantified using the ELISA-based MethylFlash Methylated DNA 5-mC Quantification Kit from EpiGentek (Farmingdale, NY, USA). hRPTCs transfected with pcDNA (control plasmid) were used as control. In accordance with the manufacturer’s guidelines, 100 ng of total genomic was used for 5-methyl cytosine quantification.

### 4.12. Methylated DNA Immunoprecipitation Assay

Methylated DNA immunoprecipitation (MeDIP) was performed, as described previously [70,71]. Briefly, genomic DNA, prepared from the hRPTCs expressing GPR37l1, was isolated using the DNeasy Blood and Tissue Kit (Qiagen, Fremont, CA, USA) and immunoprecipitated with anti-5-methyl-cytidine antibodies (EpiGentek, Farmingdale, NY, USA). The immunocomplex was digested with protease K. The immunoprecipitated DNA was cleaned using the QIAquick PCR purification kit (Qiagen, Fremont, CA, USA) and used for real-time qPCR using an ABI Prism 7900 HT (Applied Biosystems). RT2 SYBR Green ROX qPCR Master Mix (Cat. No./ID: 330524, Qiagen Inc.) was used. The PCR primers (forward: AGGTTCCTGCTGAAGACAATAG, reverse: AACCCTCCTTCATAAACGCA) used produced an amplicon of a 121 bp fragment of the NHE3 promoter region from −463 to −342 from TSS. The results were normalized to input DNA and analyzed using the ΔΔCt method [69].

### 4.13. Statistics

All results are expressed as mean ± SEM Comparisons between or among experimental groups were performed using Student’s *t*-test or one-way ANOVA, Holm–Sidak test using GraphPad Prism (GraphPad Software, La Jolla, CA, USA).

## Figures and Tables

**Figure 1 ijms-23-14456-f001:**
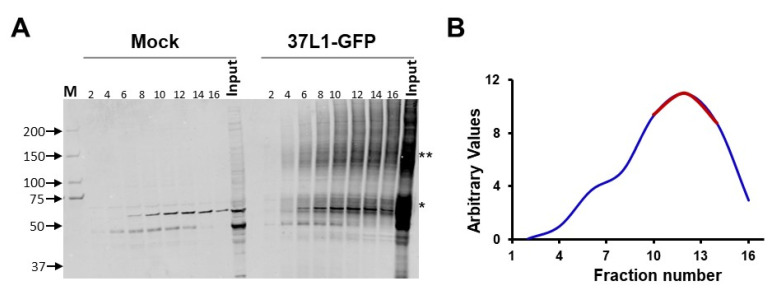
**Glycerol gradient fraction of GPR37L1 protein complex.** Protein preparations from mock- or GPR37L1-GFP-transfected hRPTCs were subjected to glycerol gradient (10–30%) centrifugation. (**A**) Fractions collected from the top of the gradient were subjected to immunoblotting for GPR37L1-GFP. Positions of the molecular size markers are indicated on the left. Positions of GPR37L1-GFP * monomer and ** high molecular aggregate are indicated on the right. (**B**) Graph showing the densitometric quantification of GPR37L1-GFP in the glycerol gradient fractions (blue). Fractions 10–14 (red) containing the highest levels of GPR37L1-GFP were pooled and used for co-immunoprecipitation (Co-IP), followed by mass spectrometry analyses.

**Figure 2 ijms-23-14456-f002:**
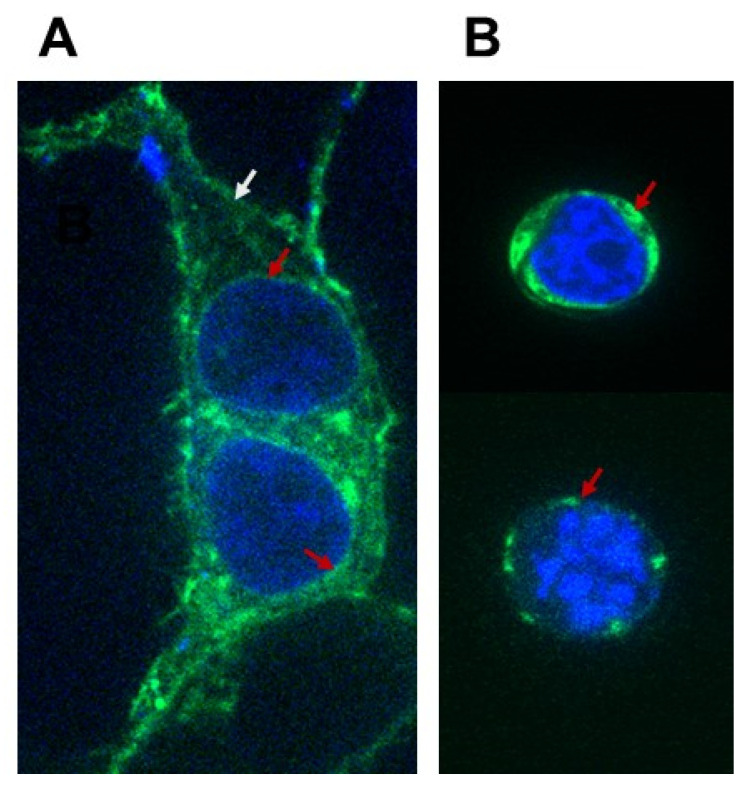
**GPR37L1-GFP localized at the plasma and the nuclear membrane.** Immunofluorescence images of whole cell and nuclei of hRPTCs showing the expression of GPR37L1-GFP at the plasma and the nuclear membrane. hRPTCs grown on glass bottom microwell dishes were transfected with the plasmid GPR37L1-GFP. Two days following transfection, the (**A**) whole cells or (**B**) nuclei prepared as described in the methods were fixed with 1% PFA for 10 min. Subcellular locations of GPR37L1-GFP expression were detected by fluorescence confocal microscopy at 600× magnification. GPR37L1-GFP, green; nucleus, blue. An example for the locations of the GPR37L1-GFP on the plasma membrane and nuclear membrane are indicated by the white and red arrows, respectively.

**Figure 3 ijms-23-14456-f003:**
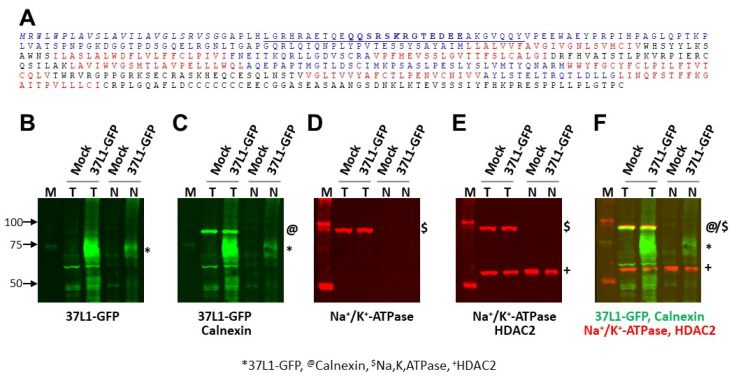
**GPR37L1 protein has a nuclear localization signal.** Nuclear localization signal (NLS) and immunoblots showing the expression of GPR37L1 in the nucleus. (**A**) Amino acid sequence of GPR37L1 showing the potential bipartite (underlined) and classical monopartite (bold) NLS sequence on the extracellular domain amino terminus, predicted using the nuclear localization signal predictor tools http://nls-mapper.iab.keio.ac.jp/cgi-bin/NLS_Mapper_form.cgi (accessed on 15 March 2019) and http://mleg.cse.sc.edu/seqNLS/ (accessed on 15 March 2019). The signal peptide sequence, the first 25-amino-acid sequence at the amino terminus, is indicated in italics. Extracellular domains are indicated in blue, transmembrane helices in red, and intracellular domains in black. (**B**–**E**) hRPTCs were transfected with a plasmid encoding GPR37L1-GFP or with a plasmid pcDNA, an empty vector (mock). (**B**) Two days after transfection, the expression of GPR37L1-GFP in total (T) or nuclear (N) protein preparations was detected by immunoblotting using an antibody specific to GFP. The same blot was probed sequentially for (**C**) calnexin, a marker for endoplasmic reticulum; (**D**) Na^+^/K^+^-ATPase, a marker for membrane proteins; or (**E**) HDAC2, a marker for nuclear proteins. IRDye^®^ 800CW-labeled donkey anti-rabbit IgG (926-32213, Licor, Lincoln, NE, USA) and goat anti-rat IgG (926-32219, Licor, Lincoln, NE) were used for GPR37L1-GFP and calnexin. IRDye^®^ 680RD-labeled donkey anti-mouse IgG (926-68022, Licor, Lincoln, NE, USA) was used for Na^+^/K^+^-ATPase and HDAC2. Total or nuclear protein prepared from the hRPTCs transfected with pcDNA (mock), an empty vector, served as control. Molecular size markers (M) are shown in each panel and the sizes of the markers in kDa are indicated to the left of panel B. Positions of the target proteins are indicated by symbols on the right side of the immunoblots: * for GPR37L1-GFP, @ for calnexin, $ for Na^+^/K^+^-ATPase, and + for HDAC2. (**F**) Panel showing the overlay of the Western blots for the four targets using two-color infrared fluorescent protein detection for GPR37L1 and calnexin (green fluorescent signals), and Na^+^/K^+^-ATPase and HDAC2 (red fluorescent signals).

**Figure 4 ijms-23-14456-f004:**
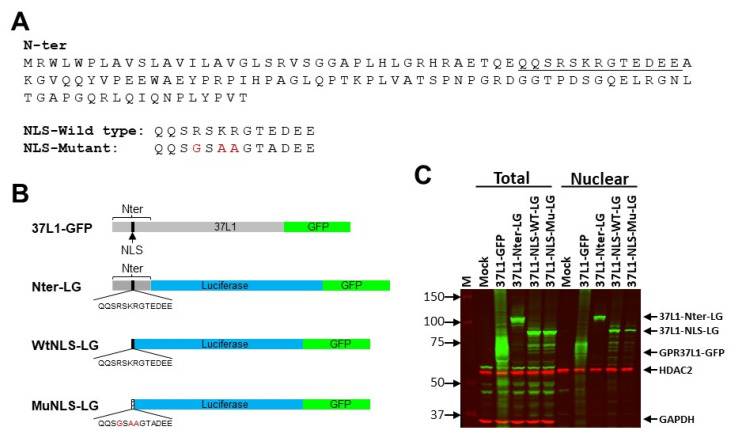
**NLS for GPR37L1 is located on the amino terminus**. (**A**) N-terminus (N-ter) 124-amino-acid sequence of GPR37L1 showing the potential monopartite (underlined) NLS sequence, NLS wild-type, and NLS mutant sequences fused to luciferase–GFP fusion (LG) constructs. (**B**) Diagram showing the plasmid constructs of wild-type or the N-terminus mutants of GPR37L1 tagged to LG fusion cDNA (LG) (37L1-Nter-LG) or the wild-type or mutant NLS tagged to LG (37L1-WtNLS-LG or 37L1-MuNLS-LG). (**C**) hRPTCs were transfected with the indicated plasmids. The wild-type or mutant GPR37L1 expressions were determined by immunoblotting of the total or nuclear protein preparations, using two-color infrared fluorescent protein detection. Expression of the LG constructs was detected using the GFP antibodies (green). The blot was probed sequentially for HDAC2 (red), a marker for nuclear protein, or GAPDH (red), a marker for cytoplasmic proteins. IRDye^®^ 800CW-labeled donkey anti-rabbit IgG (926-32213, Licor, Lincoln, NE, USA) was used for GPR37L1-GFP, and IRDye^®^ 680RD-labeled donkey anti-mouse IgG (926-68022, Licor, Lincoln, NE, USA) was used for HDAC2 and GAPDH. hRPTCs transfected with pcDNA, an empty vector, served as control. Positions of the molecular size markers are indicated on the left, and the GPR37L1-GFP, 37L1-NLS-LG, HDAC2, and GAPDH proteins are indicated on the right.

**Figure 5 ijms-23-14456-f005:**
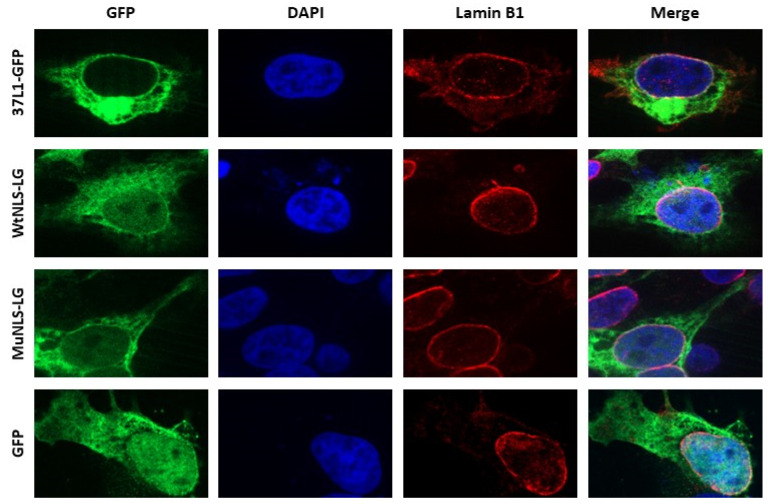
**Amino terminus NLS is required for nuclear expression of GPR37L1**. hRPTCs were transfected with the indicated GPR37L1 constructs as in Figure 4A. Two days later, the expressions of wild-type or mutant GPR37L1 proteins were examined by immunofluorescence confocal microscopy at 600× magnification. The cells were also stained for lamin B1, a marker for nuclear inner membrane, and DAPI, a marker for nucleus. GPR37L1, green; nucleus, blue; lamin B1, red; plasma membrane, purple. White dots in the merge column indicate the expression of WtNLS-LG at the nuclear envelope.

**Figure 6 ijms-23-14456-f006:**
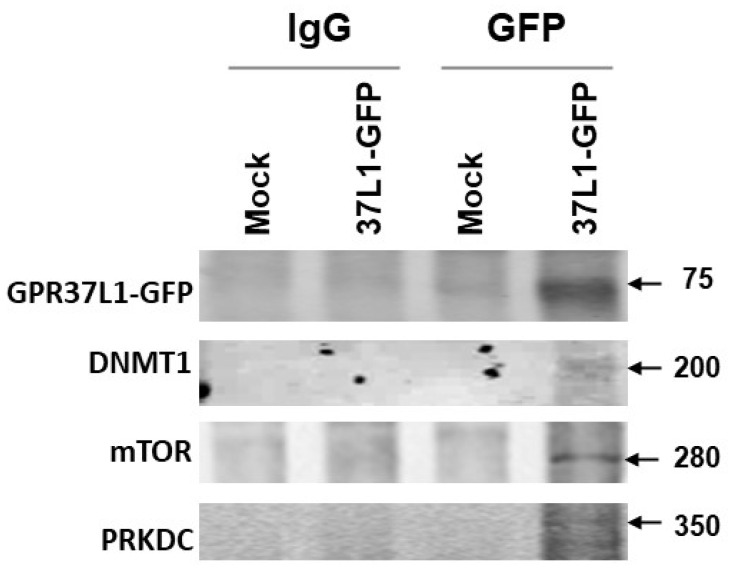
**GPR37L1 interacts with DNMT1, mTOR, and PRKDC**. The glycerol gradient fractions 10–14 of the protein preparation from hRPTCs expressing GPR37L1-GFP (37L1-GFP) or the mock-transfected hRPTCs (mock) from (Figure 1) were used for Co-IP using the GFP antibody (GFP) or control rabbit IgG (IgG). The Co-IP samples were immunoblotted to detect GPR37L1-GFP, DNMT1, mTOR, or PRKDC. The molecular size markers are indicated on the right.

**Figure 7 ijms-23-14456-f007:**
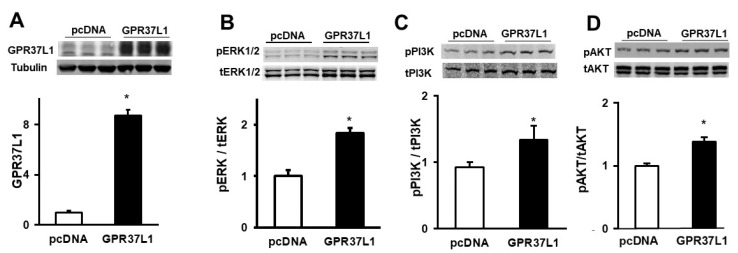
**GPR37L1 activates PI3/AKT/mTOR signaling.** PI3K/AKT/mTOR signaling components in hRPTCs expressing GPR37L1 were probed by immunoblotting. The hRPTCs were transfected with GPR37L1 (GPR37L1) or control plasmid (pcDNA). (**A**) Immunoblots of GPR37L1 in hRPTCs transfected with pGPR37L1. (**B**–**D**) Quantification of phospho-ERK1/2,(p-ERK1/2), phospho-PI3K (pPI3K), and phospho-AKT (S473) (pAKT) was normalized to the corresponding total (t) protein. *n* = 3/group, * *p* < 0.05 vs. their respective pcDNA, *t*-test.

**Figure 8 ijms-23-14456-f008:**
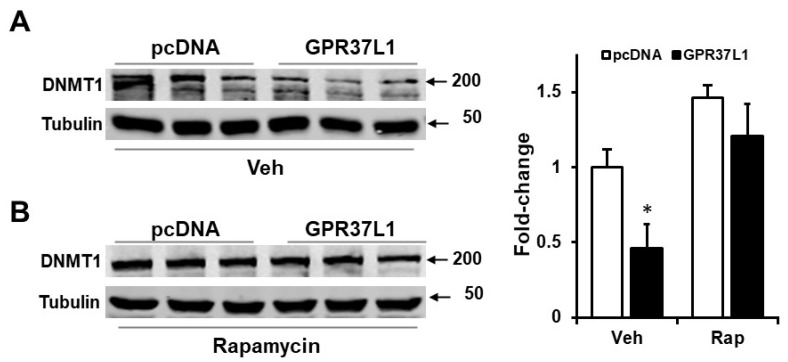
**GPR37L1 decreases DNMT1 expression through mTOR.** The hRPTCs, transfected with GPR37l1 or control plasmid (pcDNA), were treated with vehicle (Veh) or rapamycin (50 nM [Rap], for 24 h). (**A**,**B**) Protein preparations from the hRPTCs were immunoblotted for DNMT1. Quantification of DNMT1 expression normalized by tubulin. *n* = 3/group, * *p* < 0.05 vs. pcDNA+Veh, *t*-test.

**Figure 9 ijms-23-14456-f009:**
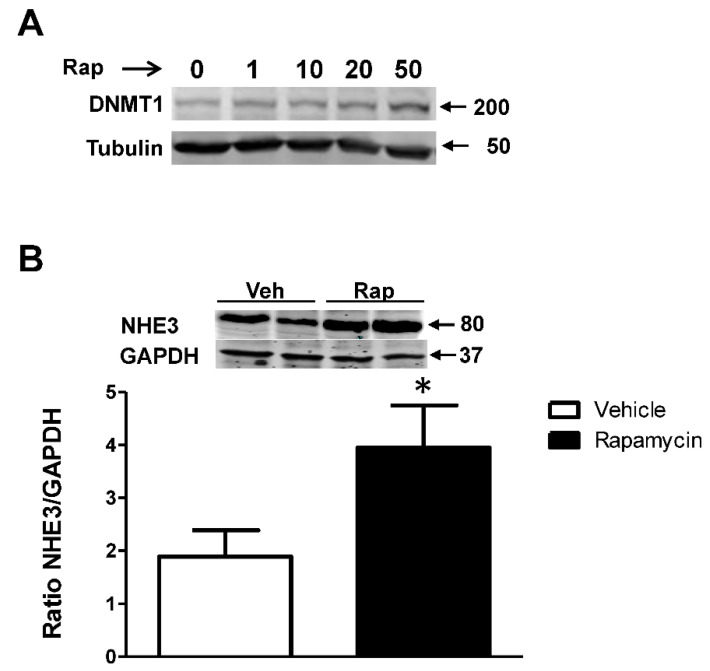
**Rapamycin treatment increases DNMT1 and NHE3 expression in hRPTCs. Cells** were treated with rapamycin at the concentrations of 1, 10, 20, and 50 nM or vehicle for 24 h. The expression of DNMT1 was determined by Western blot. Quantification was normalized by tubulin expression. The molecular size markers are indicated on the left. Another set of hRPTCs were treated for 24 h with 50 nM rapamycin or vehicle, and the expression of NHE3 of the cell lysates was determined by Western blot. Quantification was normalized by GAPDH. The molecular size markers are indicated on the left. *n* = 3/group * *p* < 0.05 vs. vehicle.

**Figure 10 ijms-23-14456-f010:**
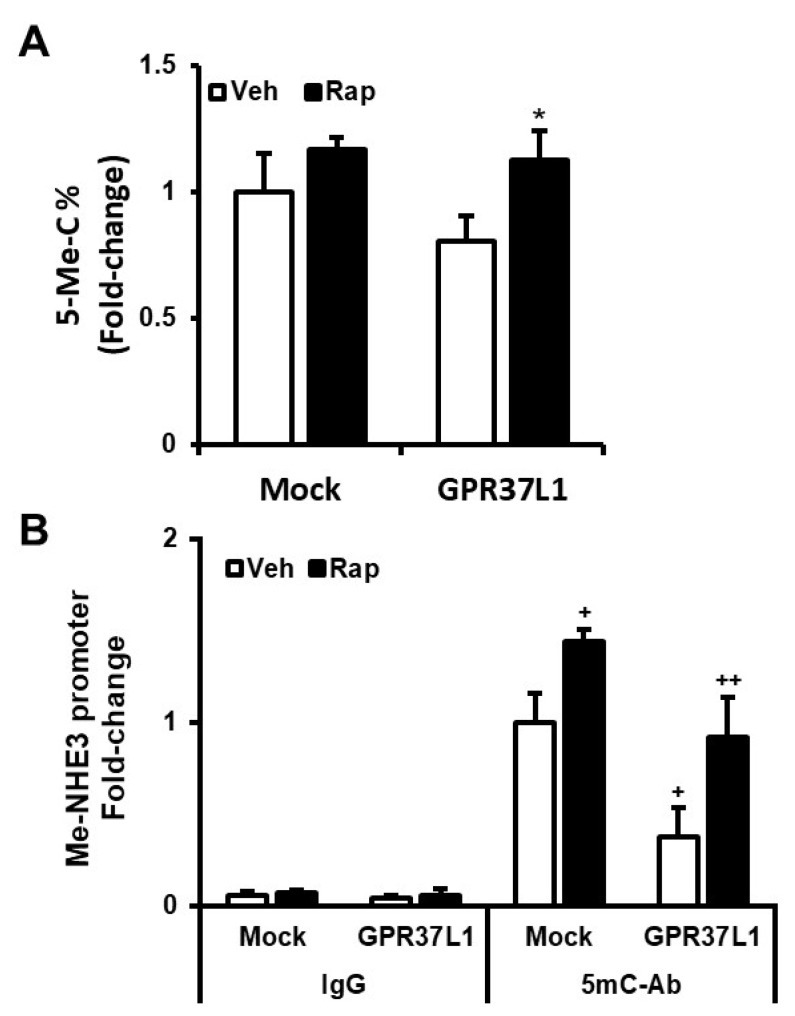
**GPR37L1 inhibits genomic DNA methylation.** (**A**) Genomic DNA prepared form hRPTCs expressing GPR37L1-GFP were analyzed for 5-cytosine methylation using EZ DNA Methylation-Gold Kit (Zymo Research, Irvine, CA, USA). *n* = 3/group, * *p* < 0.05 vs. GPR37L1/Veh, t-test. (**B**) Methylation status of NHE3 promoter region was determined by methylated DNA immunoprecipitation followed by quantitative PCR. *n* = 3/group, ^+^
*p* < 0.05 vs. 5mC-Ab/Mock/Veh, *t*-test. ^++^
*p* < 0.05 vs. 5mC-Ab/GPR37L1/Veh, *t*-test.

**Figure 11 ijms-23-14456-f011:**
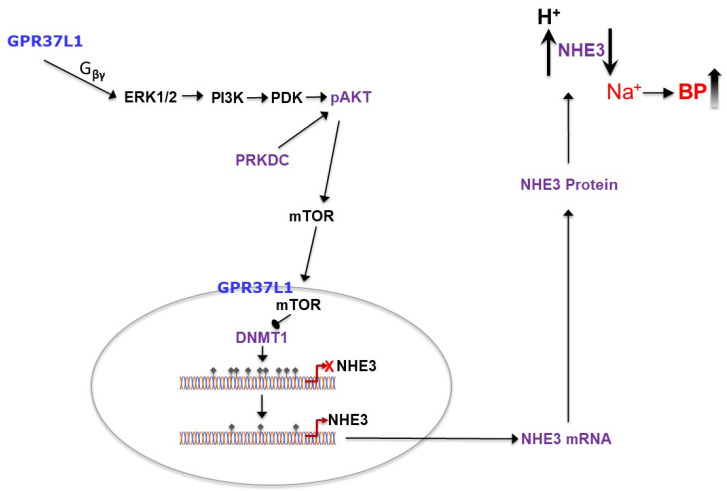
**Schematic diagram showing the GPR37L1 signaling pathway in hRPTCs**. In hRPTCs, GPR37L1 located on the nuclear membrane signals via the Gβγ to phosphorylate ERK/1/2, which in turn phosphorylates PI3K and activates AKT, a main downstream target of mTOR. Activation of mTOR inhibits DNMT1 in the nucleus and increases the transcription of NHE3 through a DNMT1-mediated decrease in the methylation of the NHE3 promoter.

**Table 1 ijms-23-14456-t001:** List of GPR37L1-associated proteins detected by mass spectrometry.

	Protein ID	Protein Names	Gene Names
Nuclear transport proteins			
	O95373	Importin-1	IPO1
	O15397	Importin-7	IPO7
	O14980	Exportin-1	XPO1
	Q9HAV4	Exportin-5	XPO5
	Q14974	Importin subunit beta-1	KPNB1
	P52292	Importin subunit alpha-1	KPNA2
	Q8TEM1	Nuclear pore membrane glycoprotein 210	NUP210
**Kinases**			
	P42345	Serine/threonine-protein kinase mTOR	MTOR
	O14654	Insulin receptor substrate 4	IRS4
	P78527	DNA-dependent protein kinase catalytic subunit	PRKDC

**DNA methylation**	P26358	DNA (cytosine-5)-methyltransferase 1	DNMT1

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
