# Peer review of "G Protein-Coupled Receptor 37L1 Modulates Epigenetic Changes in Human Renal Proximal Tubule Cells"

_ijms, 2022, doi:10.3390/ijms232214456_

Round 1
Reviewer 1 Report
This manuscript describes the role of the G-protein-coupled receptor GPR37L1 as a modulator of epigenetic changes driven by its interaction with DNMT1. The findings are novel and seem to have a functional connection to the GPR37L1-mediated changes in the expression of NHE3 in kidney.
Comments
- The manuscript seems to focus on two different aspects of GPR37L1: the localization of the receptor in the nuclear membrane and the role of the receptor in epigenetics. The major weakness in this manuscript is the apparent lack of any connection between these two aspects. The authors do not discuss this issue at all. What would be necessary is to determine if the nuclear localization of the receptor is obligatory for its role as a modulator of epigenetics.
- Another major weakness in the manuscript is the lack of direct evidence that DNA methylation mediates the changes in the expression of NHE3 driven by GPR37L1. It is convincing that GPR37L1 interacts with DNMT1 and that this interaction changes the methylation status of the NHE3 promoter. Previous studies from the same authors have shown that GPR37L1 induces the expression of NHE3. Now the question is: are these two events related? The authors need to demonstrate that the expression of NHE3 in renal proximal tubular cells is indeed controlled by DNA methylation.
- The data show that GPR37L1 localizes to the nuclear membrane and also to the plasma membrane. The data also show the involvement of the receptor in cellular signaling related to phosphorylation of AKT and PI3K. Is there any connection between the changes in these phosphorylation events and the differential localization of the receptor? In other words, is it the plasma membrane receptor or the nuclear membrane receptor that is responsible for the effects of the receptor on the phosphorylation of PI3K and AKT?
- IP in Fig. 6 show at least three different species for DNMT1, but when the authors studied the effect of GPR37L1 and rapamycin, the WB in Fig. 8 focuses on one or two specific species. What is the rationale for this? Which species is taken for quantification shown in Fig. 8?
- Fig. 7. The data for the panel D are missing. It is surprising that that the manuscript was not considered incomplete before it was sent out for review.
- There are several discrepancies in the presentation of the data. (i) In many figures, the legend for MW should mention that the numbers refer to kDa. (ii) The signal peptide in Fig. 2A is 24 amino acids, in contrast to the claim that it is 25 amino acids. (iii) Why are there differences between Fig. 1 and Fig. 2 in terms of GPR37L1 protein complexes, which are evident in Fig. 1 but not in Fig. 2? (iv) In Fig. 5, the legend mentions about the use of cholera toxin b as a marker for the plasma membrane, but there are no data in the figure. (v) There needs to be a consistency in denoting the receptor as GPR37L1 and GPR37l1; use consistently GPR37L1 for human and Gpr37l1 for mouse; (vi) In Fig. 7, the phosphoproteins seem to have been normalized to total AKT or PI3K instead of tubulin as claimed in the legend. (vii) In Fig. 8, indicate the MW for the DNMT1 species. (viii) With regard to the quantification of the data in Fig. 8, the results section states 70% reduction in DNMT1 protein, but the actual data in the figure do not seem to agree with this number. (ix) Supplement Fig. S2: the doses of rapamycin indicated in the figure and in the legend do not agree.
- The authors need to discuss as to how the NLS is involved in the localization of the receptor in the nuclear membrane. NLS is known to be needed for the import of proteins across the nuclear membrane into the nucleus. But GPR37L1 is not found within the nucleus; it is located on the membrane. How does it relate to the presence of NLS in the protein sequence?
- Fig. 10. Does the model indicate that all the signaling events related to the receptor occur because of the receptor in the plasma membrane? What is then the role of the receptor in the nuclear membrane in the biology of the receptor? This issue relates to the major weakness of the manuscript, meaning that it is not clear whether the two findings, namely the signaling events and the differential localization of the receptor, are actually related.
Author Response
Reply to the Reviewer 1 comments
Comments and Suggestions for Authors
This manuscript describes the role of the G-protein-coupled receptor GPR37L1 as a modulator of epigenetic changes driven by its interaction with DNMT1. The findings are novel and seem to have a functional connection to the GPR37L1-mediated changes in the expression of NHE3 in kidney.
Comments
- The manuscript seems to focus on two different aspects of GPR37L1: the localization of the receptor in the nuclear membrane and the role of the receptor in epigenetics. The major weakness in this manuscript is the apparent lack of any connection between these two aspects. The authors do not discuss this issue at all. What would be necessary is to determine if the nuclear localization of the receptor is obligatory for its role as a modulator of epigenetics.
We found this criticism very helpful and have almost completely re-written the manuscript to highlight our thinking of how GRP37L1 acts in both the plasma and the nuclear membrane. We have added data in Figure 6 that shows that GPR37L1 expression also increases the phosphorylation of ERK1/2. GPCRs are able to phosphorylate ERK1/2 through its association with Gβγ an effect which has been shown to other GPCRs linked to Gαi. We hope now the connection between the plasma and the nuclear membrane is clearer.
- Another major weakness in the manuscript is the lack of direct evidence that DNA methylation mediates the changes in the expression of NHE3 driven by GPR37L1. It is convincing that GPR37L1 interacts with DNMT1 and that this interaction changes the methylation status of the NHE3 promoter. Previous studies from the same authors have shown that GPR37L1 induces the expression of NHE3. Now the question is: are these two events related? The authors need to demonstrate that the expression of NHE3 in renal proximal tubular cells is indeed controlled by DNA methylation.
To address this criticism we have included a new experiment (Figure 9) in which we show that in human renal proximal tubule cells endogenously expressing GPR37L1 treatment with rapamycin indeed increases the protein expression of NHE3 .
- The data show that GPR37L1 localizes to the nuclear membrane and also to the plasma membrane. The data also show the involvement of the receptor in cellular signaling related to phosphorylation of AKT and PI3K. Is there any connection between the changes in these phosphorylation events and the differential localization of the receptor? In other words, is it the plasma membrane receptor or the nuclear membrane receptor that is responsible for the effects of the receptor on the phosphorylation of PI3K and AKT?
This concern was partly addressed in comment #1. We hypothesized that GPR37L1 is internalized
together with its signaling complex and serves as a scaffold for mTOR to act on DNMT1. Since this is
a speculation, we have not included it in the manuscript.
- IP in Fig. 6 show at least three different species for DNMT1, but when the authors studied the effect of GPR37L1 and rapamycin, the WB in Fig. 8 focuses on one or two specific species. What is the rationale for this? Which species is taken for quantification shown in Fig. 8?
The 200 kDa band for DNMT1 is the only one recognized by the Cell Signaling antibody as the
specific for DNMT1. This band was quantified in all experiments.
- Fig. 7. The data for the panel D are missing. It is surprising that that the manuscript was not considered incomplete before it was sent out for review.
We are very sorry about this mistake and is corrected in the revised version.
- There are several discrepancies in the presentation of the data. (i) In many figures, the legend for MW should mention that the numbers refer to kDa. (ii) The signal peptide in Fig. 2A is 24 amino acids, in contrast to the claim that it is 25 amino acids. (iii) Why are there differences between Fig. 1 and Fig. 2 in terms of GPR37L1 protein complexes, which are evident in Fig. 1 but not in Fig. 2? (iv) In Fig. 5, the legend mentions about the use of cholera toxin b as a marker for the plasma membrane, but there are no data in the figure. (v) There needs to be a consistency in denoting the receptor as GPR37L1 and GPR37l1; use consistently GPR37L1 for human and Gpr37l1 for mouse; (vi) In Fig. 7, the phosphoproteins seem to have been normalized to total AKT or PI3K instead of tubulin as claimed in the legend. (vii) In Fig. 8, indicate the MW for the DNMT1 species. (viii) With regard to the quantification of the data in Fig. 8, the results section states 70% reduction in DNMT1 protein, but the actual data in the figure do not seem to agree with this number. (ix) Supplement Fig. S2: the doses of rapamycin indicated in the figure and in the legend do not agree.
Issues i, ii, iv, v, vi, vii, and ix were corrected in the revised version
- The authors need to discuss as to how the NLS is involved in the localization of the receptor in the nuclear membrane. NLS is known to be needed for the import of proteins across the nuclear membrane into the nucleus. But GPR37L1 is not found within the nucleus; it is located on the membrane. How does it relate to the presence of NLS in the protein sequence?
The following sentence was included in the revised version and the corresponding reference was
included in the Reference List:
Localization to nuclear membranes can be mediated by a classical nuclear localization sequence
(NLS) and/or importins. Nuclear targeting via an NLS or importins can occur after GPCR
internalization
Crilly SE, Puthenveedu MA. Compartmentalized GPCR Signaling from Intracellular Membranes. The
Journal of Membrane Biology 2021: 254:259–271
.
- Fig. 10. Does the model indicate that all the signaling events related to the receptor occur because of the receptor in the plasma membrane? What is then the role of the receptor in the nuclear membrane in the biology of the receptor? This issue relates to the major weakness of the manuscript, meaning that it is not clear whether the two findings, namely the signaling events and the differential localization of the receptor, are actually related.
This concern has been answered in comments #1 and #3

Reviewer 2 Report
The authors show that the nuclear G-protein coupled receptor GPR37L1 activates the PI3K/Akt/mTOR pathway and that, in turn, mTOR inhibits DNMT1, thereby liberating the NHE3 gene promoter from inhibitory methylation effects. As a sodium transporter, increased NHE3 expression in the renal proximal tubule cells ostensibly increases sodium reabsorption. Thus, the authors propose a molecular mechanism for hypertension.
Minor Concerns:
1. Some typo errors, e.g. line 55 “on the role of GPR37L1…” and “form” should be “from” on line 339.
2. The sentence on line 427 is incomplete.
Major Concerns:
1. Figure 7D is missing and does not get referenced in the text, though it appears as a placeholder.
2. One important thing to consider when analyzing the effects of ectopic gene expression on cell signaling pathways is saturation of the system. Therefore, it would be good if the authors could replicate these effects on signaling with reciprocal experiments, in which they knockdown GPR37L1 and observe decreases in Akt and PI3K phosphorylation.
3. The authors demonstrate a decrease in NHE3 promoter methylation, but lean on their previous publication to show that GPR37L1 alters expression. It would be nice if the authors could verify those data and show the difference in expression of NHE3 at the mRNA and protein levels here. Potentially, this could be done through 5-Aza treatment or siRNA against DNMT1.
Author Response
Reply to the Reviewer 2 comments
Comments and Suggestions for Authors
The authors show that the nuclear G-protein coupled receptor GPR37L1 activates the PI3K/Akt/mTOR pathway and that, in turn, mTOR inhibits DNMT1, thereby liberating the NHE3 gene promoter from inhibitory methylation effects. As a sodium transporter, increased NHE3 expression in the renal proximal tubule cells ostensibly increases sodium reabsorption. Thus, the authors propose a molecular mechanism for hypertension.
Minor Concerns:
- Some typo errors, e.g. line 55 “on the role of GPR37L1…” and “form” should be “from” on line 339.
- The sentence on line 427 is incomplete.
We are very sorry about these mistakes and are corrected in the revised version.
Major Concerns:
- Figure 7D is missing and does not get referenced in the text, though it appears as a placeholder.
This was also corrected in the revised version
- One important thing to consider when analyzing the effects of ectopic gene expression on cell signaling pathways is saturation of the system. Therefore, it would be good if the authors could replicate these effects on signaling with reciprocal experiments, in which they knockdown GPR37L1 and observe decreases in Akt and PI3K phosphorylation.
We have added data in Figure 6 that shows that GPR37L1 expression also increases the phosphorylation of ERK1/2. GPCRs are able to phosphorylate ERK1/2 through its association with Gβγ an effect which has been shown to other GPCRs linked to Gαi.
- The authors demonstrate a decrease in NHE3 promoter methylation, but lean on their previous publication to show that GPR37L1 alters expression. It would be nice if the authors could verify those data and show the difference in expression of NHE3 at the mRNA and protein levels here. Potentially, this could be done through 5-Aza treatment or siRNA against DNMT1.
To address this criticism we have included a new experiment (Figure 9) in which we show that in human renal proximal tubule cells endogenously expressing GPR37L1 treatment with rapamycin indeed increases the protein expression of NHE3 .